# Reasons for intending to accept or decline kidney cancer screening: thematic analysis of free text from an online survey

Charlotte Freer-Smith,[1] Laragh Harvey-Kelly,[1] Katie Mills,[2] Hannah Harrison,[2] Sabrina H Rossi,[3] Simon J Griffin,[2] Grant D Stewart,[4] Juliet A Usher-Smith [2]

[1]School of Clinical Medicine, University of Cambridge, Cambridge, UK
[2]The Primary Care Unit, Department of Public Health and Primary Care, University of Cambridge, Cambridge, UK
[3]Department of Oncology, University of Cambridge, Cambridge, Cambridgeshire, UK
[4]Department of Surgery, University of Cambridge, Cambridge, UK

**Correspondence to**
Juliet A Usher-Smith;
jau20@medschl.cam.ac.uk

## ABSTRACT

**Objectives** Kidney cancer has been identified as a disease for which screening might provide significant benefit for patients. The aim of this study was to understand in detail the facilitators and barriers towards uptake of a future kidney cancer screening programme, and to compare these across four proposed screening modalities.

**Design** An online survey including free-text responses.

**Setting** UK

**Participants** 668 adults

**Primary and secondary outcome measures** The survey assessed participants' self-reported intention to take-up kidney cancer screening with four different test methods (urine test, blood test, ultrasound scan and low-dose CT). We conducted thematic analysis of 2559 free-text comments made within the survey using an inductive approach.

**Results** We identified five overarching themes that influenced screening intention: 'personal health beliefs', 'practicalities', 'opinions of the test', 'attitudes towards screening' and 'cancer apprehension'. Overall, participants considered the tests presented as simple to complete and the benefits of early detection to outweigh any drawbacks to screening. Dominant facilitators and barriers varied with patterns of intention to take up screening across the four tests. Most intended to take up screening by all four tests, and for these participants, screening was seen as a positive health behaviour. A significant minority were driven by practicalities and the risks of the tests offered. A smaller proportion intended to reject all forms of screening offered, often due to fear or worry about results and unnecessary medical intervention or a general negative view of screening.

**Conclusions** Most individuals would accept kidney cancer screening by any of the four test options presented because of strong positive attitudes towards screening in general and the perceived simplicity of the tests. Providing information about the rationale for screening in general and the potential benefits of early detection will be important to optimise uptake among uncertain individuals.

### Strengths and limitations of this study

► The free-text nature of the questions allowed participants to provide their own reasoning without being prompted or constrained by a list of predefined options.
► The large sample size meant that we were able to explore reasons for both intending to take up screening and for not taking up screening.
► Our use of an online recruitment platform means the views of those who completed our survey may not be representative of the general population.
► It also meant we relied on written comments from participants and were unable to explore their comments or views in-depth.
► In the absence of an existing screening programme, we were also only able to assess reasons behind intention and not attendance at screening.

common cancer in women,[1] with over 400 000 new cases being diagnosed in 2018.[2] The incidence of kidney cancer in the UK is projected to rise by 26% between 2014 and 2035, representing one of the fastest accelerating cancers within that timeframe.[3] Nearly, a quarter of individuals diagnosed with kidney cancer in the UK have evidence of metastatic disease at presentation;[4] the 5-year age-adjusted relative survival is 86% for those diagnosed with the earliest stage of the disease, but drops to 12% for metastatic disease. These epidemiologic data have led to increasing interest in screening for kidney cancer, with the development of a suitable screening programme being identified as a key research priority.[5]

For a screening programme to be successful, it must be acceptable to the general public and uptake must be high. Previous research in the context of other screening programmes has identified a number of factors influencing screening uptake. These include practical barriers, for example, difficulty in

### INTRODUCTION

Kidney cancer is the ninth most common cancer worldwide in men and the 14th most

making an appointment or lack of time to do so,[6 7] as well as psychosocial barriers such as fear or embarrassment.[7–9] Identified facilitators of screening attendance encompass both societal factors such as public education, physician recommendation and social networks,[10 11] and personal factors such as positive attitudes towards screening,[12] the desire for peace of mind[9 13] and subjective assessments of individual health status and cancer risk.[11 14] However, many of these existing studies have identified public opinion through qualitative methods with small sample sizes, or via surveys using predefined lists of possible responses. Furthermore, these studies have focused on cancers for which there are existing national screening programmes (breast, cervical and colorectal cancer). Compared with these cancers, kidney cancer has a lower prevalence,[3] public awareness of the condition is lower,[15] and the range of potential screening modalities is greater. Understanding the specific reasons why people intend to take up or decline kidney cancer screening with different screening tests is therefore important when developing future kidney cancer screening programmes.

In an online survey, we have shown that a future kidney cancer screening programme would be positively received and intention to attend high, while the choice of screening test is likely to have an important influence on uptake.[16] The aim of the current study was to use free-text responses to questions in the online survey to describe the reasons for intending to accept or decline kidney cancer screening with four different putative screening test options (urine dipstick analysis, blood based biomarkers, low dose CT and ultrasound). The free-text responses are used to understand in detail the reasons for those findings, identify the specific barriers and facilitators for kidney cancer screening and to compare these drivers across four proposed screening modalities.

## METHODS
### Study design
This study reports analysis of free-text answers from an online population-based survey of 668 members of the public in the UK aged 45–77 years.

### Participants and recruitment
Participants were recruited to complete the survey through an online participant recruitment platform for researchers (Prolific, www.prolific.ac). Individuals registered with Prolific were eligible to take part if they were between 45 and 79 years old and had a Prolific approval rating ≥95% (reflecting the percentage of studies completed that have been approved by researchers). The survey was added to the Prolific platform on 20 August 2019. At that time, 7767 met the eligibility criteria for the study. Based on demographic data available on the Prolific website, approximately 80% of these were White/Caucasian, 34% male, 53% lived in the UK, 15% were current smokers, 43% had a university level education and 51% considered themselves to be in the top five

deciles of socioeconomic status. The survey was live for 8 hours. During this time, 1190 participants viewed the participant information leaflet and consent form. A total of 1021 participants completed the survey. The analysis reported here is based on the free-text comments from the 668 UK-based participants. We chose to limit to UK participants for this analysis to remove any differences in views arising from cultural differences in attitudes towards cancer, screening and different healthcare or insurance systems.

### Survey
A copy of the relevant sections of the survey can be found in online supplemental file 1. In brief, after providing basic demographic data, participants were presented with an overview of kidney cancer, and then the details of four potential screening methods (urine testing, blood biomarkers, low-dose CT and ultrasound). This information included the number of people likely to test positive with each test method, and of those, how many people would actually have cancer. These summaries were developed in collaboration with patient and public representatives, based on the best available evidence, and the information was reviewed by two urologists within the research team (GDS and SHR). Following presentation of the details of the screening methods, participants were asked how likely they would be to take part in screening with each method (response options were 'Very likely', 'Likely', 'Unlikely' and 'Very unlikely'). Participants were subsequently asked to 'describe in a few words why?' they had given that response for each screening modality. This analysis focuses on the free-text responses to those four open-ended questions.

### Qualitative analysis
We conducted thematic analysis of all free-text comments using an inductive approach, with initial coding of responses being driven by the content of the comments themselves. Responses were analysed in two groups based on whether participants responded positively ('likely' or 'very likely' to take part) or negatively ('unlikely' or 'very unlikely' to take part). From this point, these groups will be referred to as 'likely' or 'unlikely', respectively. During the analysis process, comments that were inconsistent with the participant's previous answer were removed (eg, if they answered 'very unlikely' to take part but provided a free-text answer supporting screening). Incomplete data (where participants did not provide any free text) were also removed.

Two researchers (KM, a researcher with qualitative expertise and CF-S a medical student with a qualitative research background) familiarised themselves with a random sample of 400 comments, representing 15% of the total data and equally split between those likely or unlikely to take part, and developed initial codes independently. They then discussed and reviewed the coding decisions and developed two consolidated lists:

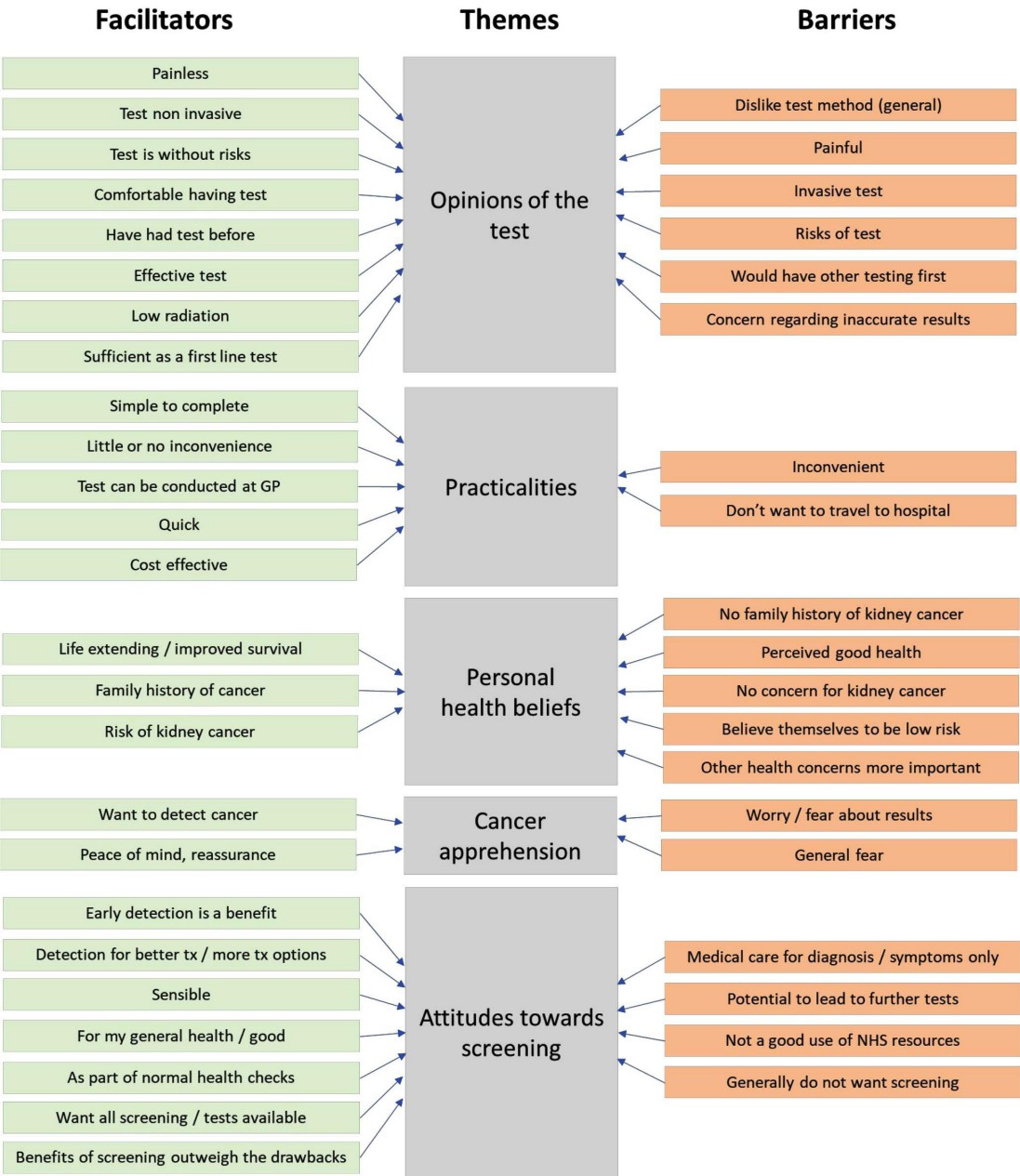

**Figure 1** Individual codes (facilitators and barriers) mapped onto overarching themes.

one for the 'likely' to take part comments, and one for the 'unlikely' to take part comments (figure 1). The remaining free-text comments were coded into these agreed lists within an excel file by one researcher (CF-S). If a comment contained multiple meanings, the comment was assigned to as many codes as appropriate to cover the content. At this stage, 5% of the total sample, selected at random, was coded by two researchers (CF-S and LH-K) and any discrepancies discussed. Agreement between the two was 94%. The full list of codes was then discussed at a meeting with the entire research team. Within this meeting, the group mapped the individual codes into overarching themes.

To compare the relative importance of individual codes and themes, the number of times a code was mentioned

for each of the four screening tests was counted. We used this data first to explore differences in responses between screening modalities, descriptively highlighting the differences in code frequencies. We then divided participants into three subgroups: those likely to take up screening by all four test methods, those unlikely to take up screening by any of the methods and those whose intention varied between the tests. Differences in the patterns of responses between these groups were explored. Finally, we investigated the patterns of responses with sex, age, level of education and social group.

**Ethics approval and consent to participate**
Ethical approval was obtained from the Psychology Research Ethics committee of the University of Cambridge

(Ref 2019.055). All participants gave informed written online consent before they began the survey.

## Patient and public involvement

Two members of the public contributed, through face-to-face discussions and email, to the development of the survey and participant information sheet. They also provided comments on the final manuscript prior to submission.

## RESULTS

The sociodemographic profile of the sample is shown in table 1. 373 (54%) were female, with a mean age of 54.7±7 years and a good distribution across age groups. Most (73%) participants were in social group ABC1 (those with managerial, administrative or professional job roles) and 288 (43.1%) had university level education.

### Intention to take up kidney cancer screening

Overall, the majority of participants (518, 78%) were likely to take up screening by all four screening test modalities. Of the remaining participants, 119 (18%) would be likely to take up some screening tests but not all, with 55/119 (46%) likely to attend for an ultrasound, urine or blood testing, but not for a low-dose CT scan. 31 (5%) indicated that they were unlikely to take up screening by any of the test methods. Since intention to take-up screening was high, we collected a greater number of free-text comments relating to reasons why someone was likely to take part in screening. Once inconsistent and blank data had been removed (51 and 62 participants, respectively), there were 2304 comments from those 'likely' to take up screening and 255 comments from those 'unlikely' to take up a screening.

### Themes identified

From the free-text comments, we identified 49 individual codes (27 facilitators and 22 barriers from those likely and unlikely to take up screening, respectively). On average, each comment was coded under 1.5 codes: 1448 (56.6%) of comments were relevant to one code, 907 (35.4%) were relevant to two codes, with the remaining 204 (8%) relevant to three or more. Together, the codes mapped onto five overarching themes (figure 1): personal health beliefs; cancer apprehension; attitudes towards screening; opinions of the test and practicalities (additionally, we included an 'other' category containing five codes that did not map elsewhere). Illustrative quotes from each theme are presented in table 2.

### Facilitators and barriers identified across screening methods

Table 3 shows the frequency with which each of the facilitators and barriers were reported, overall and across the four screening modalities. 648 participants provided comments relating to blood tests. 589 (91%) of those were likely to attend. Comments among that group focused primarily on statements that blood testing is 'simple to complete' (160, 25%), the

| Table 1 | Participant characteristics |
| --- | --- |
| | **UK (n=668)** **n (%), mean (SD)** |
| **Age** | |
| 45–49 | 188 (28) |
| 50–54 | 195 (29) |
| 55–59 | 123 (18) |
| 60–64 | 91 (14) |
| ≥65 | 71 (11) |
| Mean (SD) | 54.7 (7.0) |
| **Sex** | |
| Female | 373 (54) |
| Male | 295 (44) |
| **University education** | |
| Yes | 288 (43) |
| No | 380 (57) |
| **Ethnicity** | |
| White | 653 (98) |
| Other | 15 (2) |
| Missing | 0 |
| **General health measure** | |
| Excellent, very good, good | 524 (78) |
| Fair, poor | 144 (22) |
| **Smoking** | |
| Non-smoker | 340 (51) |
| Ex-smoker | 236 (35) |
| Current smoker | 92 (14) |
| **BMI** | |
| Mean (SD) | 27.4 (5.8) |
| Range | 16.5–50.0 |
| Missing | 29 (4) |
| **Previous diagnosis of cancer?** | |
| Yes | 34 (5) |
| No | 634 (95) |
| **Family history of kidney cancer?** | |
| Yes | 19 (3) |
| No | 635 (95) |
| Missing | 14 (2) |
| **Social group** | |
| ABC1 | 500 (75) |
| C2DE | 136 (20) |
| Missing | 32 (5) |

BMI, body mass index.

'benefits of screening outweigh the drawbacks' (84, 13%) and 'early detection is a benefit' (73, 11%). Of those unlikely to take up blood testing (59, 9%), the only consistently mentioned barrier was a 'dislike of the test method', mentioned by 3% (21).

Among the 637 participants providing comments on urine testing, 608 (95%) were likely to take

**Table 2** Illustrative quotes of themes

| Qualitative theme | Barriers: illustrative quotes | Facilitators: illustrative quotes |
|---|---|---|
| Personal health beliefs | 'I have a healthy lifestyle and at a low risk for kidney cancer'. (F, 55–59 years, ultrasound) 'There is no kidney cancer or cancer in my immediate family'. (F, 45–49 years, blood test) | 'As my parent had kidney cancer, I would very much put myself forward to screening'. (F, 60–64 years, ultrasound) 'Because life is precious, I want to live as long as possible'. (M, 55–59 years, CT) |
| Practicalities | 'Because my mobility problems make it very difficult for me to travel anywhere'. (F, 55–59 years, ultrasound) 'If it involved a specific visit, it's unlikely I would want to go'. (M, 55–59 years, urine test) | 'It's no inconvenience at all to have any of the screening methods discussed'. (M, 50–54 years, urine test) 'I have had ultrasound scans before, its quick and easy' (F, 45–49 years, ultrasound scan) |
| Opinions of the test | 'I would not wish to receive a dose of radiation to test for the extremely small chance of kidney cancer'. (M, 65–69 years, CT) 'Really don't like needles, I'd opt for something else if it were available'. (M, 45–49 years, blood tests) | 'A simple procedure and blood tests don't really alarm me at all'. (M, 70–74 years, blood test) 'Easy to do when attending the doctors for something else and not painful in anyway'. (F, 50–54 years, urine test) |
| Attitudes towards screening | 'Generally, I do not like medical appointments unless I need treatment'. (M, 55–59 years, blood tests) 'There are a lot of cancers, you can't screen for all of them. If I had symptoms, I would see a GP'. (F, 50–54 years, ultrasound) | 'Benefit of early detection outweighs minor inconvenience of screening' (M, 55–59 years, blood test) 'Would be foolish not to. If the disease has no symptoms and can be identified through screening, has the potential to save lives'. (F, 55–59 years, CT) |
| Cancer apprehension | 'I would be too afraid of the results. Short of a transplant, what could be done anyway?' (F, 55–59 years, urine test) 'The stress of going to the hospital waiting for results is too great'. (F, 60–64 years, CT) | 'Why waste a chance to "set nerves at rest"?' (M, 55–59 years, blood test) 'High value of early detection or peace of mind' (M, 50–54 years, blood test) |

F, female; M, male.

up screening. Similar to the responses for blood testing, the most common overall reason for taking up screening was that the test is 'simple to complete' (282, 44%). Frequently cited other comments that related specifically to the characteristics of the test were 'little or no inconvenience' (79, 12%) and 'quick' (57, 9%). 'Early detection is a benefit' was also frequently mentioned by this group (63, 10%) but references to 'benefits of screening outweigh the drawbacks' were less common, potentially reflecting that fewer drawbacks are associated with urine testing than blood testing. Overall, no single barrier to urine testing was identified, with no code occurring more than six times.

631 participants provided comments relating to ultrasound scans and 643 relating to CT. Intention to take up screening was again high with 575 (91%) likely to attend ultrasound and 532 (82%) CT. Notably, among participants likely to accept screening by ultrasound or CT, the most mentioned facilitator for both does not relate to the test itself but the belief that 'early detection is a benefit' (97 (15%) for CT scan, 107 (17%) for ultrasound). Although less dominant, 'simple to complete' was still mentioned for both (55 (9%) for CT, 80 (13%) for ultrasound).

For ultrasound, there are also multiple codes within the theme 'opinions of the test' that are mentioned more frequently than for other test methods: 'painless' (57, 9%) 'non-invasive' (53, 8%) and 'without risks' (46, 7%). Of these facilitators, only 'painless' is mentioned with some frequency for CT scans (45, 7%). As seen for blood tests, 'benefits of screening outweigh the drawbacks' is also a dominant facilitator for CT scans.

The key barrier identified by those unlikely to take up a CT scan is the 'risk of the test' (53, 8%), some mentioning only concerns regarding the danger of radiation, while others linked this to long term cancer risks. An additional barrier highlighted for both CT and ultrasound, but not blood or urine testing, was travel to a hospital (10 (2%) and 9 (1%) respectively).

Notably, across all test modalities and across those likely or unlikely to attend screening, there were very few comments that related to the risk of kidney cancer specifically or the accuracy or relative costs of the tests. Evaluation of the responses by demographics also revealed no clear patterns with age, sex, social group or university education.

**Table 3** Facilitators and barriers of screening uptake across modalities, frequency of mention of codes and themes

| Themes | Individual codes | Blood n=589 | Urine n=608 | CT n=532 | USS n=575 | Total n=2304 | Theme total |
|---|---|---|---|---|---|---|---|
| Personal health beliefs | Life extending/improved survival | 34 | 34 | 54 | 48 | **204** | **224** |
| | Family history of cancer | 2 | 1 | 2 | 3 | **9** | |
| | Risk of kidney cancer | 0 | 1 | 2 | 7 | **11** | |
| Practicalities | Simple to complete | 160 | 282 | 55 | 80 | **859** | **1468** |
| | Little or no inconvenience | 38 | 79 | 26 | 32 | **254** | |
| | Test can be conducted at GP | 22 | 32 | 0 | 4 | **90** | |
| | Quick | 42 | 57 | 32 | 43 | **231** | |
| | Cost effective | 8 | 10 | 4 | 2 | **34** | |
| Opinions of the test | Painless | 18 | 26 | 45 | 57 | **172** | **803** |
| | Test non-invasive | 6 | 23 | 27 | 53 | **132** | |
| | Test is without risks | 21 | 23 | 23 | 46 | **136** | |
| | Comfortable having test | 52 | 17 | 17 | 23 | **126** | |
| | Have had test before | 34 | 15 | 20 | 39 | **123** | |
| | Effective test | 25 | 24 | 14 | 13 | **100** | |
| | Low radiation | 0 | 1 | 10 | 2 | **14** | |
| | Sufficient as a first test | 8 | 9 | 1 | 2 | **29** | |
| Attitudes towards screening | Early detection is a benefit | 73 | 63 | 97 | 107 | **403** | **928** |
| | Detection for better tx/more tx options | 16 | 16 | 25 | 25 | **98** | |
| | Sensible | 42 | 44 | 54 | 73 | **257** | |
| | For my general health/good | 22 | 18 | 41 | 39 | **138** | |
| | As part of other normal health checks | 15 | 6 | 3 | 2 | **32** | |
| | Want all screening/tests available | 41 | 34 | 47 | 52 | **208** | |
| | Benefits of screening outweigh the drawbacks | 84 | 13 | 72 | 28 | **210** | |
| Cancer apprehension | Want to detect cancer | 48 | 35 | 41 | 54 | **213** | **485** |
| | Peace of mind/reassurance | 46 | 43 | 66 | 74 | **272** | |
| Other | General agreement | 32 | 22 | 25 | 15 | **116** | **129** |
| | Influence of HCP recommendation | 3 | 1 | 4 | 4 | 13 | |
| | | **Blood** | **Urine** | **CT** | **USS** | **Total** | **Theme total** |
| | | **n=59** | **n=29** | **n=111** | **n=56** | **n=255** | |
| Personal health beliefs | No family history of cancer | 1 | 1 | 1 | 1 | **5** | **50** |
| | Perceived good health | 2 | 2 | 0 | 2 | **8** | |
| | No concern for kidney cancer | 2 | 3 | 3 | 6 | **17** | |
| | Believe themselves to be low risk | 2 | 1 | 2 | 6 | **12** | |
| | Other health concerns more important | 1 | 1 | 2 | 3 | **8** | |
| Practicalities | Inconvenience | 3 | 3 | 7 | 4 | **20** | **43** |
| | Do not want to travel to hospital | 2 | 1 | 10 | 9 | **23** | |

**Table 3** Continued

| Themes | Individual codes | Blood n=589 | Urine n=608 | CT n=532 | USS n=575 | Total n=2304 | Theme total |
|---|---|---|---|---|---|---|---|
| Opinions of the test | Dislike test method (general) | 21 | 1 | 9 | 1 | 33 | 124 |
| | Painful | 3 | 0 | 0 | 1 | 4 | |
| | Invasive test | 3 | 0 | 0 | 0 | 3 | |
| | Risks of the test | 1 | 0 | 53 | 1 | 55 | |
| | Would have other testing first | 2 | 0 | 11 | 2 | 15 | |
| | Concern regarding inaccurate results | 2 | 2 | 3 | 5 | 14 | |
| Attitudes towards screening | Medical care for diagnosis/ symptoms only | 4 | 4 | 7 | 6 | 25 | 70 |
| | Potential to lead to further tests | 1 | 3 | 1 | 0 | 8 | |
| | Not a good use of NHS resources | 2 | 2 | 4 | 2 | 12 | |
| | Do not want any screening | 4 | 4 | 4 | 9 | 25 | |
| Cancer apprehension | Worry/fear about the results | 6 | 6 | 6 | 7 | 31 | 53 |
| | General fear | 5 | 2 | 7 | 6 | 22 | |
| Other | General negative comment | 4 | 0 | 5 | 3 | 12 | 36 |
| | Concerns regarding medical profession | 2 | 1 | 3 | 3 | 10 | |
| | Beliefs about treatment of kidney cancer | 4 | 2 | 4 | 2 | 14 | |

The colour gradients shown indicate where each value falls within the range of data shown calculated within either the facilitator, or the barrier data set

GP, general practitioner; HCP, healthcare professional; NHS, National Health Service; tx, treatment.; USS, ultrasound scan.

## Subgroup thematic analysis

Comparison of the comments made by participants who said they were likely to take up screening by all four test methods with those who would accept screening by some of the test methods only is presented in table 4. For those whose likelihood of taking up screening varied between the different tests, 'practicalities' and 'opinions of the test' remained the most prominent themes. 'Simple to complete' represented the most frequently mentioned individual code (41%, n=118/290 comments), with most of these from comments in relation to urine or blood testing. The non-invasive nature of ultrasound scanning, and perceived lack of risks associated with this test were also seen in this group.

Conversely, for participants who would accept screening with all four tests, the themes 'attitudes towards screening', 'cancer apprehension' and 'personal health beliefs' are proportionally more important. While 'simple to complete' is still the most mentioned individual code, others within these additional themes become equally important. The facilitators identified within the 'attitudes towards screening' theme reflect a general attitude that screening is a 'correct' form of behaviour: 'sensible' (10%, n=200/2014 comments), 'for my general health/ good' (6%, n=115/2014 comments), 'early detection is a benefit' (16%, n=323/2014 comments). The perceived benefits of early detection that were highlighted by participants included improved survival, better treatment and more treatment options. Many participants, however, simply stated that early detection is always beneficial and did not give further explanation. Another common belief described by this group was the idea that screening could provide a sense of 'peace of mind/reassurance' that they did not have the disease (11%, n=219/2014), highlighting one way in which cancer worry can motivate screening attendance.

Table 5 shows the barriers cited by those who are unlikely to take up screening by any of the tests and those whose attitude towards screening varied with screening modality. For those declining some test methods, but not all, the only dominant theme is 'opinions of the test': most of these comments reference the 'risks of the test' for CT scan (54%, 45/84), or a 'dislike of the test' for blood tests (59%, 19/32). Very few other barriers outside of these are mentioned. In contrast, those who decline all screening methods do not mention test specific concerns, instead 'attitudes towards screening', 'personal health beliefs' and 'cancer apprehension' are all more frequently raised across all the test modalities. The most frequently mentioned individual barrier to screening is 'worry/fear about results' (17%, 18/109). Many also state that they 'do not want any screening' (15%, 16/109) or

**Table 4** Subgroup analysis: participants likely to attend screening, frequency of mention of facilitators

| Themes | Individual codes | Participants likely to attend all screening modalities | | | | | | Participants likely to attend screening by some, but not all screening modalities | | | | | |
|---|---|---|---|---|---|---|---|---|---|---|---|---|---|
| | | Blood n=510 | Urine n=499 | CT n=510 | USS n=495 | Total n=2014 | Theme total | Blood n=79 | Urine n=109 | CT n=22 | USS n=80 | Total n=290 | Theme total |
| Personal health beliefs | Life extending/improved survival | 30 | 33 | 54 | 46 | 163 | 180 | 4 | 1 | 0 | 2 | 7 | 8 |
| | Family history of cancer | 2 | 1 | 2 | 2 | 7 | | 0 | 0 | 0 | 1 | 1 | |
| | Risk of kidney cancer | 0 | 1 | 2 | 7 | 10 | | 0 | 0 | 0 | 0 | 0 | |
| Practicalities | Simple to complete | 131 | 216 | 51 | 61 | 459 | 793 | 29 | 66 | 4 | 19 | 118 | 215 |
| | Little or no inconvenience | 32 | 64 | 23 | 25 | 144 | | 6 | 15 | 3 | 7 | 31 | |
| | Test can be conducted at GP | 13 | 22 | 0 | 2 | 37 | | 9 | 10 | 0 | 2 | 21 | |
| | Quick | 33 | 40 | 30 | 31 | 134 | | 9 | 17 | 2 | 12 | 40 | |
| | Cost effective | 6 | 7 | 4 | 2 | 19 | | 2 | 3 | 0 | 0 | 5 | |
| Opinions of the test | Painless | 15 | 21 | 42 | 46 | 124 | 541 | 3 | 5 | 3 | 11 | 22 | 133 |
| | Test non-invasive | 6 | 16 | 24 | 35 | 81 | | 0 | 7 | 3 | 18 | 28 | |
| | Test is without risks | 16 | 14 | 22 | 27 | 79 | | 5 | 9 | 1 | 19 | 34 | |
| | Comfortable having test | 45 | 16 | 16 | 19 | 96 | | 7 | 1 | 1 | 4 | 13 | |
| | Have had test before | 26 | 12 | 20 | 36 | 94 | | 8 | 3 | 0 | 3 | 14 | |
| | Effective test | 17 | 14 | 14 | 10 | 55 | | 8 | 10 | 0 | 3 | 21 | |
| | Low radiation | 0 | 1 | 10 | 1 | 12 | | 0 | 0 | 0 | 1 | 1 | |
| | Sufficient as a first test | 6 | 7 | 1 | 1 | 15 | | 2 | 2 | 0 | 1 | 5 | |
| Attitudes towards screening | Early detection is a benefit | 70 | 61 | 94 | 98 | 323 | 734 | 3 | 2 | 3 | 9 | 17 | 47 |
| | Detection for better tx/more tx options | 15 | 15 | 25 | 23 | 78 | | 1 | 1 | 0 | 2 | 4 | |
| | Sensible | 40 | 41 | 51 | 68 | 200 | | 2 | 3 | 3 | 5 | 13 | |
| | For my general health/good | 21 | 17 | 41 | 36 | 115 | | 1 | 1 | 0 | 3 | 5 | |
| | As part of other normal health checks | 9 | 4 | 3 | 2 | 18 | | 6 | 2 | 0 | 0 | 8 | |
| | Want all screening/tests available | 41 | 32 | 45 | 48 | 166 | | 0 | 2 | 2 | 4 | 8 | |
| | Benefits of screening outweigh the drawbacks | 74 | 13 | 68 | 25 | 180 | | 10 | 0 | 4 | 3 | 17 | |
| Cancer apprehension | Want to detect cancer | 46 | 32 | 39 | 47 | 164 | 383 | 2 | 3 | 2 | 7 | 14 | 24 |
| | Peace of mind/reassurance | 45 | 40 | 66 | 68 | 219 | | 1 | 3 | 0 | 6 | 10 | |
| Other | General agreement | 27 | 18 | 23 | 13 | 81 | 93 | 5 | 4 | 2 | 2 | 13 | 13 |
| | Influence of HCP recommendation | 3 | 1 | 4 | 4 | 12 | | 0 | 0 | 0 | 0 | 0 | |

The colour gradients shown indicate where each value falls within the range of data shown, calculated within each subgroup

HCP, healthcare professional; tx, treatment; USS, ultrasound scan.

**Table 5** Subgroup analysis: participants unlikely to attend screening, frequency of mention of barriers

| Themes | Individual codes | Participants unlikely to attend any screening modalities | | | | | | Participants unlikely to attend screening by some, but not all screening modalities | | | | | |
|---|---|---|---|---|---|---|---|---|---|---|---|---|---|
| | | Blood n=27 | Urine n=27 | CT n=27 | USS n=28 | Total n=109 | Theme total | Blood n=32 | Urine n=2 | CT n=84 | USS n=28 | Total n=146 | Theme total |
| Personal health beliefs | No family history of cancer | 1 | 1 | 1 | 1 | 4 | 30 | 0 | 0 | 0 | 0 | 0 | 12 |
| | Perceived good health | 2 | 2 | 0 | 2 | 6 | | 0 | 0 | 0 | 0 | 0 | |
| | No concern for kidney cancer | 2 | 3 | 2 | 2 | 9 | | 0 | 0 | 1 | 4 | 5 | |
| | Believe themselves to be low risk | 1 | 1 | 0 | 3 | 5 | | 1 | 0 | 2 | 3 | 6 | |
| | Other health concerns more important | 1 | 1 | 1 | 3 | 6 | | 0 | 0 | 1 | 0 | 1 | |
| Practicalities | Inconvenience | 0 | 2 | 2 | 0 | 4 | 8 | 3 | 1 | 5 | 4 | 13 | 31 |
| | Do not want to travel to hospital | 1 | 1 | 1 | 1 | 4 | | 1 | 0 | 9 | 8 | 18 | |
| Opinions of the test | Dislike test method (general) | 2 | 1 | 1 | 0 | 4 | 15 | 19 | 0 | 8 | 1 | 28 | 102 |
| | Painful | 0 | 0 | 0 | 0 | 0 | | 3 | 0 | 0 | 1 | 4 | |
| | Invasive test | 0 | 0 | 0 | 0 | 0 | | 3 | 0 | 0 | 0 | 3 | |
| | Risks of the test | 1 | 0 | 4 | 0 | 5 | | 0 | 0 | 45 | 1 | 46 | |
| | Would have other testing first | 0 | 0 | 0 | 0 | 0 | | 2 | 0 | 11 | 2 | 15 | |
| | Concern regarding inaccurate results | 1 | 2 | 1 | 2 | 6 | | 1 | 0 | 2 | 3 | 6 | |
| Attitudes towards screening | Medical care for diagnosis/symptoms only | 4 | 4 | 3 | 4 | 15 | 37 | 0 | 0 | 4 | 2 | 6 | 20 |
| | Potential to lead to further tests | 0 | 2 | 0 | 0 | 2 | | 1 | 1 | 1 | 0 | 3 | |
| | Not a good use of NHS resources | 1 | 2 | 1 | 0 | 4 | | 1 | 0 | 3 | 2 | 6 | |
| | Do not want any screening | 4 | 4 | 3 | 5 | 16 | | 0 | 0 | 1 | 4 | 5 | |
| Cancer apprehension | Worry/fear about the results | 3 | 6 | 4 | 5 | 18 | 26 | 3 | 0 | 2 | 2 | 7 | 19 |
| | General fear | 3 | 2 | 1 | 2 | 8 | | 2 | 0 | 6 | 4 | 12 | |
| Other | General negative comment | 4 | 0 | 4 | 2 | 10 | 28 | 0 | 0 | 1 | 1 | 2 | 5 |
| | Concerns regarding medical profession | 1 | 1 | 2 | 3 | 7 | | 1 | 0 | 1 | 0 | 2 | |
| | Beliefs about treatment of kidney cancer | 4 | 2 | 3 | 2 | 11 | | 0 | 0 | 1 | 0 | 1 | |

The colour gradients shown indicate where each value falls within the range of data shown, calculated within each subgroup.
NHS, National Health Service; USS, ultrasound scan.

they believe that that 'medical care is for diagnosis/symptoms only' (14%, 15/109), reflecting a disagreement with the principle of screening in general.

## DISCUSSION
### Key findings
Through a thematic analysis of over 2000 free-text comments from an online survey, we have identified, and ranked the relative importance of, facilitators and barriers influencing potential uptake of kidney cancer screening by four possible screening modalities in a UK population. We found that, overall, across all four tests most participants considered screening simple to complete and that the benefits of early detection outweigh any burdens or harms. Urine testing was viewed as the simplest and least invasive to complete; without the risks or inconvenience associated with other tests. Additionally, we found that the dominant drivers and barriers varied with patterns of intention to take up screening across the four tests. The most frequent pattern, and therefore the largest group of individuals, was those who were likely to take up screening with all four tests. Among this group, screening was seen as a positive health behaviour, with many benefits: the tests are seen as simple, they see early detection as an advantage, or seek the peace of mind and clarity that accompanies a negative screening result. The second group was those for whom the specific test modality influences decision making. Within this group, a significant minority would attend all tests except for a CT scan, due to concern about the risk associated with a CT. Blood tests also polarised participants: most were comfortable with this method and it was seen as simple by many, but a small minority expressed a specific dislike of blood tests. Having to travel to hospital for a CT or USS also put some people off these modalities. The smallest group was those who reject all forms of screening. For these individuals, the drawbacks of screening in general are highlighted rather than opinions related to specific tests. These drawbacks include fear or worry about results and unnecessary medical intervention and are not balanced by sufficient benefits. Notably, largely absent from the reasons that participants did or did not intend to take up screening among any of these groups were comments related to the accuracy of the tests or the risk of kidney cancer specifically, despite information on both of these being included within the survey.

### Comparison with existing literature
Our finding that the decision to take up screening in this study for many of the participants was driven by the perceived value of early detection, with screening embodying a positive health behaviour (or a sensible decision) is consistent with previous studies exploring facilitators of screening in existing cancer screening programmes[10 11 13 17–19] and the view that screening is 'almost always a good idea'.[20] Our study builds on these findings by showing that these beliefs about screening extend to hypothetical new screening programmes and, for the individuals who hold these views, the nature of the screening test is not a key factor in their decision of whether to take up screening or not. This observation, that screening decisions are often driven by emotions and strongly held contextual beliefs, was also seen among those who did not intend to attend screening with any of the tests. While a small number of participants in this group appear to have been consciously weighing up the potential benefits and harms of kidney cancer screening within the context of their own personal health beliefs, our findings suggest many may be driven by a more general aversion to the concept of screening or apprehension about cancer in general or the results. This has been reported in the context of other screening programmes[12 21–25] and highlights the need for information materials to clarify the purpose of screening and advantages of early diagnosis in asymptomatic individuals[22] and to specifically address negative beliefs around the disease and cancer worry. The preference for simple and convenient tests, seen through the responses for urine and blood testing in particular in this study, has also been reported previously in the context of other cancer screening programmes.[7 14 26] Our findings additionally show that the risks associated with exposure to radiation for CT are a key barrier for some individuals.

### Strengths and limitations
The main strengths of this study are the large sample size and the free-text nature of the questions that allowed participants to provide their own reasoning without being prompted by a list of predefined options.

The large sample size meant that we were able to explore reasons for not taking up screening among the minority of participants who would be unlikely to take it up. Uptake rates for UK cancer screening programmes are consistently below 75%,[27–29] therefore, capturing the perspectives of those who are less likely to accept screening (and difficult to engage in research) is important. Since we were also only able to assess the reasoning behind the participants' intention to attend screening, and not their actual attendance, this highlights a limitation of our study, as intention may be higher than attendance. We also relied on the written comments and were unable to go back to individuals for clarification or further in-depth exploration of their views.

As a result of our use of an online recruitment method, the views of those who completed our survey may also not be representative of the general population.[30] In particular, the distribution across social groups was different: 73% of our participants reported being in the upper half of the social grades

(ABC1), whereas census data from 2011 suggest 53% of the UK population lies within this category. Given that participation in screening programmes is disproportionately lower among lower social grades,[19 31] the participants in our study may, therefore, be more enthusiastic about screening than the UK population as a whole. National screening programmes for cancer are also established within the UK and provided at no cost to participants within a state funded healthcare system. Awareness of these government programmes and the absence of cost as a barrier may have influenced the participants views. There are also known cultural variations in attitudes towards cancer and awareness of cancer risk that may influence response to screening.[32 33] While we would expect the range of attitudes to be similar in other countries, the relative frequency of the individual codes and themes may therefore differ. With online data collection, there is also the risk that participants did not concentrate and their responses do not accurately reflect their views. We sought to limit this by including an instructional manipulation check early on the questionnaire. Only 51 of the 2559 comments (1.9%) were also inconsistent, suggesting that the overwhelming majority of participants were appropriately engaged with the survey.

The free-text nature of the questions meant that participants were free to express their thoughts in their own words. This has the advantage of not constraining them by predefined lists. However, the question we asked ('Please describe in a few words why?') did not instruct participants to list all the reasons behind their choice, meaning responses ranged from one word to several sentences. Our findings, therefore, reflect the dominant views of participants when faced with the option of a given screening test, not all their views. The relative frequency of the different codes and themes reflects their importance and not the necessarily the number of participants for whom that factor may contribute to their decision.

## CONCLUSIONS

Our findings show that most individuals in a UK population think they are likely to attend kidney cancer screening by any of the four test options presented. These individuals are driven both by a general perception of screening as a good thing and the view that the tests are simple, making them likely to participate in any future kidney cancer screening programme largely without consideration of the nature of the test involved or their own risk of kidney cancer. For the significant minority for whom the practicalities and risks of the tests are important, optimising the logistics of the screening programme and ensuring that the written information is understandable, for example around the radiation risk associated with CT, will be important. Providing sufficient information to enable individuals to make an informed choice about screening will also be important for the small proportion of the population who do not intend to take up screening with any of the test options. Our findings suggest that that information should not be limited to the specific details of kidney cancer screening but should also include an explanation of the rationale for screening in general and the potential benefits of early detection.

**Acknowledgements** We would like to thank our two patient and public representatives Phil Alsop and Philip Dondi for their input and advice throughout this study.

**Contributors** JU-S, LH-K, KM, HH, SHR, GDS and SJG were involved in the design of the study. JU-S completed data collection. CF-S, KM and LH-K were involved in initial data analysis, and all authors contributed to the final analysis and interpretation of the data. CF-S and JU-S wrote the first draft of the manuscript. All authors critically reviewed the manuscript and have approved the final version.

**Funding** This work was funded by a research grant from Kidney Cancer UK (no award number). JU-S and KM were supported by a Cancer Research UK Cancer Prevention Fellowship (C55650/A21464). SHR is supported by The Urology Foundation (no award number) and a Cancer Research UK Clinical Research Fellowship (no award number). GDS is funded by the Cancer Research UK Cambridge Cancer Centre (Major Centre Award A25117) and the Renal Cancer Research Fund (no award number). HH is supported by a National Institute of Health Research Methods Fellowship (RM-SR-2017-09-009). The University of Cambridge has received salary support in respect of SJG from the NHS in the East of England through the Clinical Academic Reserve.The views expressed in this publication are those of the author(s) and not necessarily those of the NHS, the National Institute for Health Research or the Department of Health and Social Care.

**Competing interests** All authors have completed the Unified Competing Interest form at www.icmje.org/coi_disclosure.pdf (available on request from the corresponding author). GDS has received educational grants from Pfizer, AstraZeneca and Intuitive Surgical; consultancy fees from Pfizer, Merck, EUSA Pharma and CMR Surgical; Travel expenses from Pfizer and Speaker fees from Pfizer. All other authors declare that (1) they have no support from or relationships with companies that might have an interest in the submitted work in the previous 3 years; (2) their spouses, partners or children have no financial relationships that may be relevant to the submitted work and (3) they have no non-financial interests that may be relevant to the submitted work.

**Patient consent for publication** Not required.

**Provenance and peer review** Not commissioned; externally peer reviewed.

**Data availability statement** Data are available in a public, open access repository. The data set supporting the conclusions of this article are available in the University of Cambridge data repository (https://doi.org/10.17863/CAM.66416) and will be available for at least 10 years from the last access. All the data are be stored in accordance with the Data Protection Act 1998.

**ORCID iD**
Juliet A Usher-Smith http://orcid.org/0000-0002-8501-2531

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
