## [Reviewer comments · BMJ Open]

ARTICLE DETAILS

TITLE (PROVISIONAL)	Reasons for intending to accept or decline kidney cancer screening: thematic analysis of free text from an online survey
AUTHORS	Freer-Smith, Charlotte; Harvey-Kelly, Laragh; Mills, Katie; Harrison, Hannah; Rossi, Sabrina; Griffin, Simon; Stewart, Grant; Usher-Smith, Juliet

VERSION 1 – REVIEW

REVIEWER	Haewon Kim Seoul national university, college of nursing
REVIEW RETURNED	01-Dec-2020

GENERAL COMMENTS	In the absence of kidney cancer screening programme, it is helpful to plan future's screening provision against kidney cancer. With the large sample, participants voices could be collected via online platform, whereas it could interfere with the participants' in-depth perceptions or awareness toward screening uptake. Thus it is strength and limitation at the same time. For example, the facilitators and barriers could be formulated based on the previous screening related research findings (including other cancer screening), In my opinion, among the theme analysis in this paper, the most important asking / or intentions was established, then the main results could be much narrowed and elaborated than this extensive results. Despite of themes are not much creative, the paper keep the scientific integrity entirely to be published. It was great work.
---

REVIEWER	Cristiane Bergerot Instituto Unity de Ensino e Pesquisa, Centro de Câncer de Brasília, Brazil
REVIEW RETURNED	Thank you very much for allowing me to review this very interesting paper by Freer-Smith et al. The screening of facilitators and barriers towards up-take of a future kidney cancer screening program is a very relevant topic. Congratulations also for recruiting such a large sample of participants. I have some comments: 1. This study was conducted in UK but the authors do not discuss how unique aspects of the culture may have influenced the study and results.2. I believe the authors should discuss how these results could be extrapolated to other centers in other countries with different characteristics than their own. Once again thank you so much for allowing me to review this manuscript.

VERSION 1 – AUTHOR RESPONSE

Reviewer: 1

Dr. Hae Won Kim, Seoul national university

Comments to the Author:

In the absence of kidney cancer screening programme, it is helpful to plan future's screening provision against kidney cancer.

With the large sample, participants voices could be collected via online platform, whereas it could interfere with the participants' in-depth perceptions or awareness toward screening uptake. Thus it is strength and limitation at the same time. For example, the facilitators and barriers could be formulated based on the previous screening related research findings (including other cancer screening),

In my opinion, among the theme analysis in this paper, the most important asking / or intentions was established, then the main results could be much narrowed and elaborated than this extensive results.

Despite of themes are not much creative, the paper keep the scientific integrity entirely to be published.

It was great work.

We thank the reviewer for her comments and are pleased that she judged the work positively and supports the scientific integrity the study.

With regards to her comments on the online survey methodology, we agree that this represents both a strength and a limitation. The main strength was that we were able to reach a large sample size. As the reviewer highlights though a limitation was our inability to probe participants more deeply about their perceptions and opinions. We included mention of this to some extent in the strengths and limitations, highlighting that our findings represent participants dominant views, and not all their views. In light of the reviewer's comments we have expanded this section and added the text in bold below within the Strengths and Limitations section of our Discussion, page 18:

“The large sample size meant that we were able to explore reasons for not taking up screening among the minority of participants who would be unlikely to take it up. Uptake rates for UK cancer screening programmes are consistently below 75% 26–28, therefore, capturing the perspectives of those who are less likely to accept screening (and difficult to engage in research) is important. Since we were also only able to assess the reasoning behind the participants' intention to attend screening, and not their actual attendance, this highlights a limitation of our study, as intention may be higher than attendance. **We also relied on the written comments and were unable to go back to individuals for clarification or further in-depth exploration of their views.**”

We have also included this as a limitation in the Strengths and Limitations summary points, page 4:

“It also meant that we relied on written comments from participants and were unable to explore their comments or views in-depth.”

With regards to her suggestion that we narrow the main results and focus on only those themes which were most important / dominant, we strongly believe it is important to retain all

the themes and codes identified. We took this approach for two reasons. Firstly, we felt that although it was important to analyse the most dominant themes emerging, it was also important to show the range and diversity of views provided, and not sacrifice the granularity of the data. Indeed, one of the strengths of this study is the ability to not only describe but also to rank the frequency of attitudes towards screening. Secondly, a strength of this study was its large sample size, enabling us to gather feedback from participants who are unlikely to attend any screening. As these participants represent only a small proportion of the population, had we focussed only on the most important themes identified, we would have risked losing important, though less frequently mentioned, participant perspectives from this key group. We do, however, agree with the reviewer that narrowing the full lists of barriers / facilitators identified provides a useful overview for readers. It is for this reason that we identified the overarching themes and included these both in the results tables and as headings within the text.

Reviewer: 2

Dr. Cristiane Bergerot, City of Hope Comprehensive Cancer Center Duarte

Comments to the Author:

Thank you very much for allowing me to review this very interesting paper by Freer-Smith et al. The screening of facilitators and barriers towards up-take of a future kidney cancer screening program is a very relevant topic. Congratulations also for recruiting such a large sample of participants.

I have some comments:

1. This study was conducted in UK but the authors do not discuss how unique aspects of the culture may have influenced the study and results.
2. I believe the authors should discuss how these results could be extrapolated to other centers in other countries with different characteristics than their own.

We thank the reviewer for her comments and are pleased that she feels this topic is both relevant and interesting.

We agree with the reviewer's comments that unique aspects of culture within the UK are likely to have influenced the results of our study, and that this is very important when considering the interpretation of the results.

To reflect these considerations in the manuscript we have made the following additions (shown in bold), in the Methods, Discussion, Strengths and Limitations and the Conclusion:

Methods, page 7

"The analysis reported here is based on the free-text comments from the 668 UK based participants. We chose to limit to the UK participants for this analysis to remove any differences in **views arising from cultural differences in attitudes towards cancer, screening, and different** healthcare or insurance systems.

Results, page 13

"Notably, across all test modalities and across those likely or unlikely to attend screening, there were very few comments that related to the risk of kidney cancer specifically or the accuracy **or relative costs** of the tests"

Discussion, page 15 / 16

"Through a thematic analysis of over 2000 free text comments from an online survey, we have identified, and ranked the relative importance of, facilitators and barriers influencing potential uptake of kidney cancer screening by four possible screening modalities **in a UK population**.

Strengths and limitations, page 18 / 19

“Only 51 of the 2,559 comments (1.9%) were also inconsistent, suggesting that the overwhelming majority of participants were appropriately engaged with the survey. **National screening programmes for cancer are also established within the UK and provided at no cost to participants within a state funded healthcare system. Awareness of these government programmes and the absence of cost as a barrier may have influenced the participants views. There are also known cultural variations in attitudes towards cancer and awareness of cancer risk that may influence response to screening (32,33). While we would expect the range of attitudes to be similar in other countries, the relative frequency of the individual codes and themes may therefore differ..**”

Conclusion, page 19

“Our findings show that most individuals **in the UK population** think they are likely to attend kidney cancer screening by any of the four test options presented. These individuals are driven both by a general perception of screening as a good thing and the view that the tests are simple, making them likely to participate in any future kidney cancer screening programme largely without consideration of the nature of the test involved or their own risk of kidney cancer.”